# Neoadjuvant Chemo-Radiation Using IGRT in Patients with Locally Advanced Gastric Cancer

**Jing Shen** , **Xin Lian, Qiu Guan, Lei He, Fuquan Zhang and Jie Shen ***

Department of Radiation Oncology, Peking Union Medical College Hospital, Chinese Academy of Medical Sciences & Peking Union Medical College, NO.1 Shuaifuyuan Wangfujing, Dongcheng District, Beijing 100730, China
* Correspondence: 13521039164@163.com

**Abstract:** The goal of this study was to see how effective and safe neoadjuvant chemoradiation with image-guided IMRT was in patients with locally advanced resectable gastric cancer. Between January 2013 and June 2019, patients with locally advanced (cT3/cT4 or N+) gastric cancer treated with neoadjuvant chemoradiotherapy at PUMCH (Peking Union Medical College Hospital) were retrospectively studied. Using concurrent chemotherapy (Capecitabine alone or XELOX*2 cycles), radiotherapy (IMRT (intensity-modulated radiation therapy) 45 Gy, 25#, 5 weeks) was delivered with IGRT (image-guided radiotherapy) before the start of each weeks therapy to ensure accuracy and repeatability. A total of 95 patients were enrolled in the study, 93 (97.9%) stage cT3/T4 and 85 (89.5%) stage N+. Of these, 85 patients (89.5%) had a tumor located in the upper 1/3 of the stomach, and 93/95 patients (97.9%) completed neoadjuvant chemoradiation, with 80 patients (84.2%) undergoing stomach resection (58 D2 and 22 D1 gastrostomies). Pathology downstaging was found in 68 patients (85.0%), with 66 patients (82.5%) receiving T downstaging and 56 patients (70.0%) receiving N downstaging. There were 11 individuals (13.8%) who had a pathological complete response (PCR). The average period of follow-up was 44.7 months (19–96 months). The 5-year OS (overall survival), LRFS (local recurrence-free survival), and DMFS (distant metastasis free survival) rates of patients were 47.0% (95% CI: 38.6–55.4), 86.55% (95% CI: 79.1–93.99) and 60.71% (95% CI: 51.49–69.93%), respectively. Thirteen (13.7%) patients had grade 3–4 leukopenia, anemia, and thrombocytopenia, while 9 (9.5%) patients had grade 3–4 anemia, and 5 (5.3%) patients had grade 3–4 thrombocytopenia. PCR was found to be a significant predictive factor for OS in multivariate analysis (HR = 11.211, 95% CI: 1.500–83.813, $p$ = 0.024). The method of using IGRT image-guided IMRT (45 Gy, 25 fractions, 5 weeks) combined with concurrent chemotherapy in patients with locally advanced resectable gastric cancer was equally effective when compared to the clinical efficacy of neoadjuvant chemoradiotherapy, with clinical outcomes achieving equal efficacy, with similar PCR rates and high rates of OS, LRFS, and DMFS, as well as good tolerances of concurrent chemoradiotherapy with acceptable side effects.

**Keywords:** gastric cancer; preoperative chemo-radiotherapy; pathological complete response; radiotherapy

## 1. Introduction

Gastric cancer is the fifth most common cancer and the third greatest cause of cancer deaths worldwide, with China accounting for nearly half of all cases [1].

Due to the lack of typical clinical premonitory symptoms, more than 75 percent of newly diagnosed patients are in an advanced stage (the cancer has invaded the muscle layer or lymph nodes), and the survival rates of advanced-stage patients are only 20–50% [2], and approximately 50% of patients have a disease that is too advanced for surgery. However, as more chemotherapy and radiation techniques are incorporated into treatment regimens, the overall incidence and fatality rate of stomach cancer is decreasing [3].

In other gastrointestinal malignancies, such as esophageal and rectal cancers, neoadjuvant and adjuvant therapy are generally accepted to improve disease-free survival (DFS) and overall survival (OS) [4,5]. Moreover, prospective data on preoperative therapy for patients with locally advanced gastric cancer (LAGC) is scarce [6]. Several RCTs and meta-analyses have shown that neoadjuvant chemotherapy, in addition to surgery alone, improves survival [7]. Previous studies, on the other hand, have been unable to reach a firm conclusion about the best neoadjuvant therapy plan for LAGC patients. The goal of this study is to see if preoperative chemoradiation is effective and feasible in these patients.

## 2. Patients and Methods

### 2.1. Patients

This was a retrospective analysis with data collected from January 2013 and June 2019. The eligibility criteria were as follows: histologically confirmed gastric cancer with adenocarcinoma, 18–70 years old, performance status (PS) of 0–1 by Eastern Cooperative Oncology Group (ECOG) criteria, local advanced stage (cT3-4N0-2M0 or cT1-4N1-2M0) by chest and abdomen computed tomography (CT), and trans-esophageal ultrasound (some patients received positron emission tomography/computed tomography (PET/CT)), lesion located in the upper 1/3 of the stomach (below the esophagogastric junction and originates from the area connecting the cardia to the upper 1/3 of the lesser curvature of the stomach and the upper 1/3 of the greater curvature of the stomach) or the esophagogastric junction (Siewert II or Siewert III). Exclusion criteria included distal gastric body lesions, Siewert I EGJ, M1, peritoneal carcinomatosis (gross or microscopic), distant lymph node metastasis (supraclavicular or retroperitoneal), or uncontrolled medical conditions.

The study protocol is listed below, see details in Figure 1.

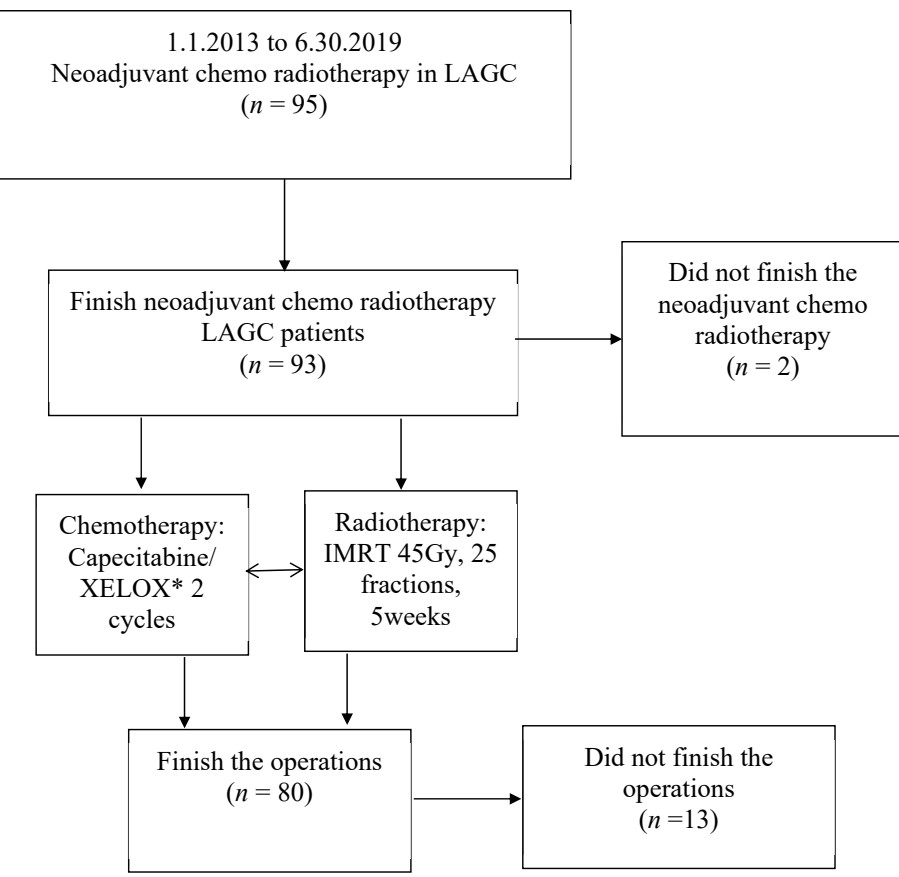

**Figure 1.** Flow diagram of study. Capecitabine: 60 mg/m$^2$ orally on days 1–5 during the radiation treatment. * XELOX: capecitabine (60 mg/m$^2$ orally on days 1–14 of a 3-week cycle) and oxaliplatin (100 mg/m$^2$, intravenously, on day 1).

### 2.2. Radiotherapy

All patients received a CT simulation (16-slice Philips Brilliance CT BigBore, Deventer, The Netherlands) in the supine position with oral and intravenous contrast agents. Bowel preparations were conducted before the CT scan (200 mL liquid to fill the stomach, e.g., water or milk). The clinical target volume (CTV) were contoured on the axial CT slices. The CT scanning range were delineated based on the endoscopy, and CT/MRI or ultrasound results taken into consideration. Involved lymph nodes were defined as short (diameter: >1 cm), or confirmed by diffusion weighted imaging or PET/CT. Gross tumor volumes (GTV) had to be delineated for the primary tumor as well as for the involved lymph nodes. The global clinical target volume (CTV) was calculated by combining the following structures: GTV, GTV lymph node and lymphatic drainage area (lower esophagus, perigastric (D1) and D2 stations), CTV tumor (which was obtained by adding a margin of 1.5 cm to GTV tumor, 0.5 cm to GTV nodal), and the lymphatic spread. The planning clinical target volume (PTV) was the CTV plus 8 mm margin in the craniocaudal direction, and 6 mm in the anteroposterior and left–right directions. The regime consists of a total dose to PTV of 45 Gy in 25 daily fractions of 1.8 Gy five days a week. Radiotherapy plans were generated on the Eclipse treatment planning system (Eclipse Inc., Madison, WI, USA). The planning goals were delivering at least 95% of the prescribed dose to 95% of the PTV. Dose prescription and recording complied with the recommendations of the ICRU 50/62 (International Commission on Radiation Units and Measurements, ICRU) [8]. Daily patient set-up was performed using laser alignment to reference marks on the skin of the patient. CBCT was used for image guidance before each day's treatment delivery. Using soft tissue registration, if the filling of the stomach did not meet the image positioning requirements, it was necessary to suspend the current treatment and restart the treatment after meeting the accuracy of the location. Patients were repositioned after co-registration of CBCT images with the planning CBCT images (see Figure 2 as an example).

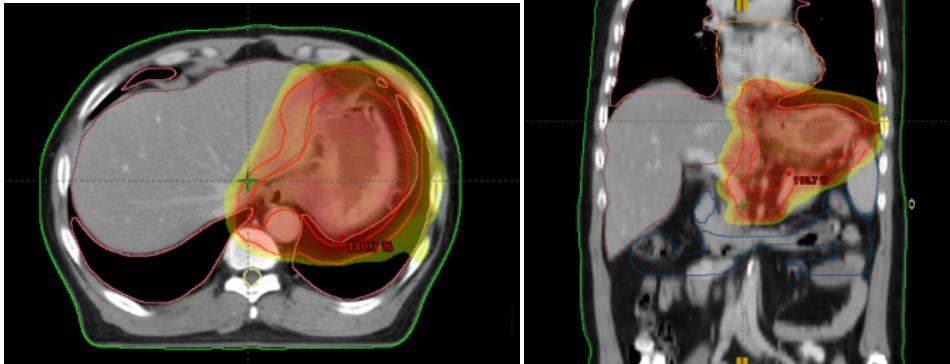

**Figure 2.** Example of image-guided radiotherapy.

### 2.3. Chemotherapy

All patients received concurrent capecitabine alone (1000 mg/m$^2$ orally on days 1–5 during the radiation treatment) or XELOX (capecitabine (1000 mg/m$^2$ by oral on days 1–14 of a 3-week cycle) and oxaliplatin (100 mg/m$^2$, intravenously, on day 1)). The choice of chemotherapy was based on the patient's age, physical condition, and economic situation. Patients were assessed for acute chemotherapy-related toxicity during therapy. Postoperative chemotherapy was capecitabine alone (1000 mg/m$^2$ by oral on days 1–14 of a 3-week cycle) for 6–8 circles after surgery.

### 2.4. Surgery

Before surgery, patients received a gastric CT/MRI/trans-esophageal ultrasound for reassessment of staging and resectability by surgeons. Gastrectomy surgery was performed for gastric carcinoma patients at least 6 weeks after neoadjuvant radiotherapy. Whether to

perform D2 gastrectomy surgery or not was decided by the attending surgeon, based on the clinical response to neoadjuvant treatment, and the patient's preference.

### 2.5. Tumor Response and Toxicity Criteria

After chemoradiation, an abdominal CT scan was performed to evaluate tumor response, according to the Response Evaluation Criteria for Solid Tumors (RECIST) 1.1.14. Pathological complete response (PCR) was defined as an absence of carcinoma cells in the primary site and lymph nodes, and pathological partial response (PPR) was defined as less than 10% residual carcinoma cells in the lesion. Adverse events were assessed according to the Common Toxicity Criteria of the National Cancer Institute (CTCAE) 4.0.

### 2.6. Follow-Up and Evaluation of Toxicities

Patients had follow-up examinations every 3 months during the first 2 years, every 6 months during the next 3–5 years, and then once each year. Carbohydrate antigen 199 and carcinoembryonic antigen levels were measured every 3 months together with imaging examinations that included CT scans of the thorax and abdomen. Chemoradiotherapy-related toxicities and postoperative complications were recorded.

Acute toxicities during chemoradiotherapy were evaluated every week. Toxicities were evaluated with Common Terminology Criteria for Adverse Events, version 4.0.

### 2.7. Statistics

The PCR rate, the clinical endpoints (including resection rate, downstaging rate, acute and postoperative complications, pattern of failure), and survival were also calculated. The 5-year OS, LRFS, and DMFS were estimated using the Kaplan–Meier method, and the univariate log rank test was used to evaluate the significance of prognostic factors for survival. Multivariate analysis, using the Cox proportional regression method, was performed for the covariates selected in the univariate analysis. An equivalent dose of 2 Gy fractions (EQD2) was calculated with $\alpha/\beta = 10$ for the tumor. A significance level of 0.05 was used. All the statistical analyses were performed using SPSS 25.0 (SPSS Inc., Chicago, IL, USA).

## 3. Results

### 3.1. Patients' Characteristics

A total of 95 patients were enrolled. Patients' and tumors' characteristics are detailed in Table 1. Of these patients, 54 (56.8%) were more than 60 years old. The majority of patients were male 81 (85.3%), all patients were diagnosed pathologically with locally advanced gastric carcinoma, 93 patients (97.9%) with stage cT3 or cT4, and 85 patients (89.5%) had positive lymph nodes. The location of tumors were in the upper 1/3 in 85 patients (89.5%). Poorly differentiated tumors accounted for more than other types of tumor (64; 67.4%). CEA (carcinoembryonic antigen) was the most related bold tumor marker, with 35 (36.8%) abnormality. A total of 71 (74.7%) patients received capecitabine alone, while 24 (25.3%) received XELOX for chemotherapy, Postoperative chemotherapy was given to 41 (51.3%) patients with capecitabine alone until one year after surgery.

### 3.2. Treatment and Acute Toxicity

A dose of 45 Gy in 25 fractions was delivered to all patients. The median interval to finish radiotherapy was 39 days (34–49 days). Accompanied with concurrent chemotherapy, capecitabine alone was used in 74.7% (71/95) of patients and XELOX in 25.3% (24/95) of patients. Grades 3–4 leukopenia, anemia, and thrombocytopenia were observed in 13 (13.7%) patients, 9 (9.5%) patients, and 5 (5.3%) patients, respectively. Seven patients (7.4%) developed grade 3 nausea.

**Table 1.** Patients' and tumors' characteristics.

| Characteristics | n | Percentage (%) |
|---|---|---|
| Age | | |
| <60 | 41 | 43.16 |
| ≥60 | 54 | 56.84 |
| Gender | | |
| Male | 81 | 85.26 |
| Female | 14 | 14.74 |
| ECOG performance status | | |
| 0 | 77 | 81.06 |
| 1 | 18 | 18.94 |
| Tumor location | | |
| Upper 1/3 | 85 | 89.47 |
| Other | 10 | 10.53 |
| Tumor differentiation | | |
| Well differentiated | 3 | 3.16 |
| Moderately differentiated | 21 | 22.11 |
| Poorly differentiated | 64 | 67.36 |
| Others | 7 | 7.37 |
| Blood tumor markers abnormal | | |
| Carcinoembryonic antigen | 35 | 36.84 |
| CA199 | 29 | 30.53 |
| CA242 | 29 | 30.53 |
| CA724 | 19 | 20.00 |
| CA125 | 14 | 14.74 |
| CA153 | 4 | 4.21 |
| Pretreatment tumor stage | | |
| T2 | 2 | 2.1 |
| T3 | 20 | 21.06 |
| T4 | 73 | 76.84 |
| Pretreatment node status | | |
| N0 | 10 | 10.53 |
| N1 | 34 | 35.79 |
| N2 | 49 | 51.57 |
| N3 | 2 | 2.11 |
| Chemotherapy regiment | | |
| Capecitabine | 71 | 74.74 |
| XELOX | 24 | 25.26 |
| Postoperative chemotherapy | | |
| Yes | 41 | 51.25 |
| No | 39 | 48.75 |

Chemoradiotherapy was finished by 93 (97.9%) of the 95 patients. Two patients did not finish the treatment (one presented with intestinal obstruction during treatment; one patient presented with intestinal hemorrhage). Eighty (84.2%) patients underwent gastrectomy: 58 patients (72.5%) underwent a D2 gastrectomy, and 22 patients (27.5%) underwent a D1 gastrectomy. Gastrectomy was not performed in the remaining 13 patients due to their late stage (4 with peritoneal carcinomatosis, 3 with liver metastasis, 3 with lung metastasis, and 3 with duodenal and pancreatic invasion).

### 3.3. Surgery and Postoperative Complications

In the 80 patients who underwent gastrectomy, pathologic CR was found in eleven patients. The primary tumor after surgery was T1 in 3 patients (3.8%), T2 in 5 patients (18.8%), T3 in 39 patients (48.8%), and T4a in 12 patients (1%). Additionally, 39 patients (48.8%) had N0, 29 patients (36.2%) had N1, 8 patients (10.0%) had N2, and 4 patients (5.0%)

had N3. The median number of dissected nodes was 19 (6–40). The median number of positive nodes was 5 (0–9).

Eight of the 80 patients had postoperative problems, including 3 who developed anastomotic fistulas, 2 who had stomach infections, 2 who had intestinal obstructions, and 1 who developed anastomotic stenosis.

Downstaging was observed in 68 patients (85.0%), including 66 patients (82.5%) with T downstaging and 56 patients (70%) with N downstaging. PCR was observed in 11 patients (13.8%).

Compared with capecitabine alone, XELOX presented no significant difference in PCR (12.3% vs. 14.3%, *p* = 0.385). Pathologic T0 was also found in 14 patients (17.5%). Of the 80 patients with positive lymph nodes before treatment, negative lymph node involvement was observed in 39 patients (48.8%).

### 3.4. Pattern of Failure and Survival

The median follow-up was 44.7 months (19–96 months). Treatment failure was experienced by 37 patients (46.3%), including local failure in 7 patients (8.8%), implant metastasis in 12 patients (15.0%), and distant metastasis in 33 patients (41.3%). Distant metastasis was the main pattern of failure, with the most common metastases being liver, followed by lung and bone. See details of failure patterns in Table 2 and Figure 3.

**Table 2.** Patterns of failure.

| Failure Sites | No. of Patients | % of Recurrence Patients (*n* = 37) |
|---|---|---|
| Single site | | |
| Local recurrence | 3 | 6.81 |
| Implant metastasis | 4 | 9.12 |
| Distant metastasis | 27 | 61.42 |
| Two sites | | |
| LR + IM | 0 | 0 |
| LR + DM | 2 | 4.51 |
| IM + DM | 6 | 13.63 |
| Three sites | 2 | 4.51 |

LR = Local recurrence, IM = Implant metastasis, DM = Distant metastasis.

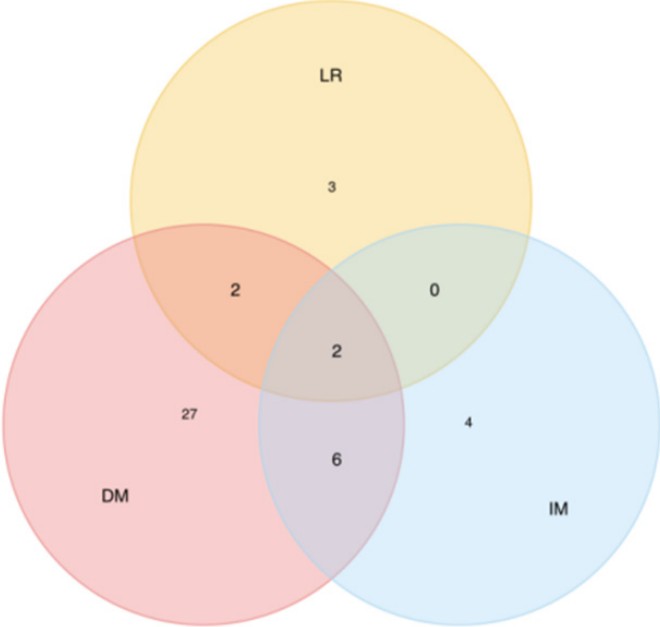

**Figure 3.** Venn diagram of patterns of failure.

The estimated 5-year OS, LRFS, and DMFS rates of patients were 46.98% (95% CI: 38.6–55.4%), 86.55% (95% CI: 79.1–91.0%), and 60.7% (95% CI: 51.5–69.9%), respectively.

Univariate analysis showed that age > 60 years, weight loss, anemia before treatment, abnormal blood tumor markers, and post-operation chemotherapy were not significant prognostic factors of OS, while chemotherapy with XELOX (comparing to capecitabine alone) ($p$ = 0.002), D2 surgery ($p$ = 0.001), PCR ($p$ = 0.003), T downstaging ($p$ = 0.001) and N downstaging ($p$ = 0.005) were significant prognostic factors of OS. Meanwhile chemotherapy with XELOX ($p$ = 0.031) was a significant prognostic factor of LC, and D2 surgery ($p$ = 0.001), PCR ($p$ = 0.025), T downstaging ($p$ = 0.001), and N downstaging ($p$ = 0.003) were significant prognostic factors of DFS.

Multivariate analysis demonstrated that PCR was a significant prognostic factor for OS (HR = 11.211, 95% CI: 1.500–83.813, $p$ = 0.024). See details in Table 3 and Figure 4.

**Table 3.** Univariate and multivariate analysis of prognostic factors in 5-year OS.

| Characteristics | Univariate Analysis | | Multivariate Analysis | | |
|---|---|---|---|---|---|
| | **Percent** | ***p*-Value** | **HR** | **95% CI** | ***p*-Value** |
| Age | | | | | |
| ≥60 | 43.90% | 0.672 | | | |
| <60 | 40.70% | | | | |
| Weight loss | | | | | |
| Yes | 38.70% | 0.849 | | | |
| No | 43.80% | | | | |
| Anemia | | | | | |
| Yes | 39.20% | 0.645 | | | |
| No | 43.80% | | | | |
| Post-operation chemotherapy | | | | | |
| Yes | 45.60% | 0.732 | | | |
| No | 30.70% | | | | |
| Chemotherapy | | | | | |
| capecitabine | 20.80% | 0.002 * | 0.286 | 0.149–0.549 | 0.148 |
| XELOX | 49.30% | | | | |
| Surgery | | | | | |
| D1 | 13.60% | 0.001 * | 3.53 | 1.866–6.679 | 0.338 |
| D2 | 56.90% | | | | |
| PCR | | | | | |
| Yes | 90.90% | 0.003 * | 11.211 | 1.500–83.813 | 0.024 * |
| No | 35.70% | | | | |
| T downstaging | | | | | |
| Yes | 52.90% | 0.001 * | 2.808 | 1.352–5.832 | 0.151 |
| No | 12.00% | | | | |
| N downstaging | | | | | |
| Yes | 50.80% | 0.005 * | 4.505 | 1.777–11.419 | 0.25 |
| No | 26.50% | | | | |

* $p$ < 0.05, significant prognostic factor.

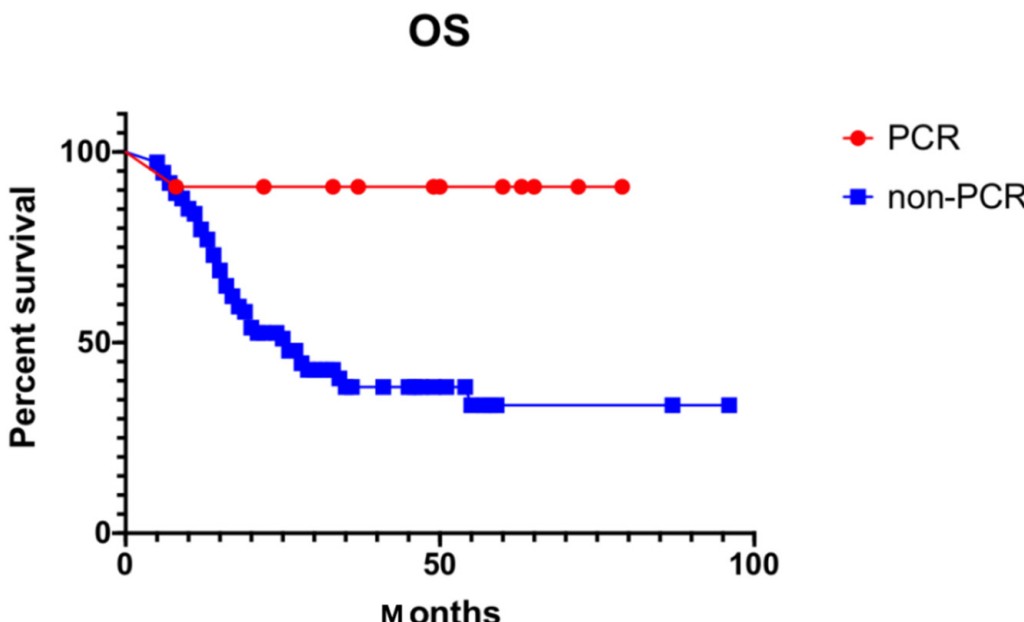

**Figure 4.** Kaplan–Meier estimate of OS of PCR (OS: overall survival; PCR: pathology complete response).

In our study, 5-year OS was 72.7% for patients who showed PCR, but only 41.0% in patients who did not ($p = 0.003$). As for whether the patients proceeded to complete adjuvant post-operative chemotherapy ($n = 41$) or not ($n = 39$) was insignificant in terms of overall survival (45.6% vs. 30.7%, $p = 0.732$).

## 4. Discussion

Gastric cancer is the most prevalent digestive system tumor, accounting for the third-most cancer deaths and fifth-most malignancies worldwide [9]. Patients with LAGC are more common in China. As more than half of patients with gastric cancer are stage II-III, the key challenge for researchers is to enhance the efficacy and survival rate of patients with LAGC [10].

In LAGC, neoadjuvant radiation therapy is utilized to minimize tumor burden, limit micrometastasis, and increase the likelihood of successful surgery, with results from phase II clinical trials MAGIC and FLOT4-AIO already showing these benefits [10]. Furthermore, the pathological response to the neoadjuvant treatment is a crucial predictive indicator for patient survival. Several clinical investigations have also proven that preoperative neoadjuvant radiation improves the therapeutic outcomes of gastric cancer patients [7,11–14] (Table 4). Phase III clinical investigations that will further compare the effect of neoadjuvant chemoradiotherapy are currently ongoing [15,16]; we look forward to the final results. In our study, the patients that received neoadjuvant chemoradiotherapy showed good tolerance, with 93 of the 95 patients completing the neoadjuvant chemoradiotherapy treatment procedure, with only 7 patients (7.4%) showing grade 3 nausea, and no grade 4 or more toxic side reactions. Moreover, the expected 5-year OS, LRFS, and DMFS were 47.0% (95% CI: 38.6–55.4%), 86.6% (95% CI: 79.1–94.0%), and 60.7% (95% CI: 51.5–69.9%), respectively. Within the limitations of a retrospective analysis and its associated bias, and in comparison with the literature reports of neoadjuvant chemotherapy in LAGC, the results of our study appear comparable to the outcomes of large randomized trials investigating neoadjuvant therapy for GC [17].

**Table 4.** Literature review of the effect of new-adjuvant radiotherapy.

| Study | Year, Country | Phase | Sample Size | Tumor Site | Groups | Local Control | Survival |
|---|---|---|---|---|---|---|---|
| Zhang et al. [11] | 1998, China | III | 370 | EGJ | RT + S vs. S | 61.4% vs. 51.7% | 10-year OS 20.3% vs. 13.3% |
| Stahl et al. [12,13] | 2009, 2019, Germany | III | 119 | EGJ | CRT + S vs. C + S | PCR 15.6% vs. 2.0% | 3-year OS 46.7% vs. 26.1%, 5-year OS 39.5% vs. 24.4% |
| Van Hagen et al. [7] | 2012, England | III | 366 | EGJ or EC | CRT + S vs. S | LRR 14% vs. 34% | 5-year OS 47% vs. 34% |
| G A von Döbeln [14] | 2019, Sweden and Norway | II | 181 | EGJ or EC | CRT + S vs. C + S | PCR 28% vs. 9% | 5-year OS 42.2% vs. 39.6% |
| Trevor Leong et al. [15] | 2019, Australia, Europa, Canada | III | 752 | EGJ or EC | CRT + S vs. C + S | On going | On going |
| Liu, X et al. [16] | 2019, China | III | 682 | EGJ or EC | CRT + S vs. C + S | On going | On going |

The PCR rate after neoadjuvant chemotherapy/chemoradiotherapy is a clear indicator of survival prognosis [18–21]. Petrelli F et al. reported that with neoadjuvant chemoradiotherapy, the PCR ratio was increased by 2.8 times (95% CI 2.27–3.47; $p < 0.001$) relative to the neoadjuvant chemotherapy [22]. In our study, the PCR of patients with neoadjuvant chemoradiotherapy was 13.8%, which was similar to the PCR rate reported by the relevant literature (13.0% to 17.0%) [12,13,22]. In this study, 13 patients who completed neoadjuvant chemoradiotherapy were found to have a new M1 in the preoperative evaluation, as well as peritoneal carcinomatosis (4), lung metastases (3), unresectable liver metastasis (3), and unresectable pancreatic invasion (3), so they did not undergo surgery due to the limited surgical treatment options and small benefits [3,23]. The median time of PFS has been obtained for 13 months, and the median survival time (MST) was 21 months. The patient did not appear more than grade 4 or more toxic side reactions. Grades 3–4 leukopenia, anemia, and thrombocytopenia were observed in 13 (13.7%) patients, 9 (9.5%) patients, and 5 (5.3%) patients, respectively. Seven patients (7.4%) developed grade 3 nausea. For the non-curable metastases in stage IV gastric cancer patients, the reported MST of conversion therapy was 6 months [24,25]. At present, digestive system tumors require a positive multidisciplinary treatment model that includes the treatment of, for example, rectum, pancreatic, and other tumors, for which radiotherapy has played a good perioperative treatment [5,26]. Radiotherapy might be one of the most significant treatment strategies in stage IV GC patients.

There are some drawbacks to this study. First, because this is a retrospective, single-center study, there is the possibility of selection bias. Second, the findings were compared to earlier literature publications, suggesting that the effects of CCRT may have been exaggerated. Despite these limitations, this is a large-scale population study exploring the efficacy of CCRT and its effects on LAGC patients; thus it has high value as a reference and it offers guidance in selecting treatment for LAGC patients.

## 5. Conclusions

Compared with the previous literature, results of preoperative neoadjuvant chemotherapy for patients with gastric cancer, the application of image-guided IMRT (45 Gy, 25 fractions, 5 weeks) combined with chemotherapy in preoperative neoadjuvant therapy for patients with locally advanced gastric cancer can achieve improved clinical efficacy, with higher rates of OS, LRFS, and DMFS, and good tolerance of concurrent chemoradiotherapy with acceptable side effects.

**Author Contributions:** Conceptualization, J.S. (Jing Shen) and X.L.; methodology, J.S. (Jie Shen); formal analysis, J.S. (Jing Shen); investigation, J.S. (Jing Shen); resources, Q.G. and F.Z.; data curation, L.H.; writing—original draft preparation, J.S. (Jing Shen); writing—review and editing, J.S. (Jie Shen); supervision, Q.G. All authors have read and agreed to the published version of the manuscript.

**Funding:** This study was funded by the National Key Research and Development Plan, the Ministry of Science and Technology of the People's Republic of China [grant number 2016YFC0105207].

**Institutional Review Board Statement:** The study was conducted in accordance with the Declaration of Helsinki, and approved by the Institutional Review Board of Peking Union Medical College Hospital (PUMCH) (protocol code K22C1342).

**Informed Consent Statement:** This is a retrospective study and written informed consent was unnecessary.

**Data Availability Statement:** Not applicable.

**Conflicts of Interest:** The authors declare no conflict of interest.

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
