# Peer review of "Neoadjuvant Chemo-Radiation Using IGRT in Patients with Locally Advanced Gastric Cancer"

_curroncol, doi:10.3390/curroncol29100586_

Round 1
Reviewer 1 Report
With interest I have read the manuscript entitled “Neoadjuvant chemoradiation using IGRT in patients with locally advanced gastric cancer”. Thank you for giving me a chance to review this article.
The paper is based on a retrospective analysis over a 6 years period. The strategy of preoperative therapy for gastric cancer has been a well-accepted research subject. At the beginning of this century and neoadjuvant chemoradiotherapy in the treatment of gastric cancer was researched in several oncological centers. Despite promising
results including good tolerance of the treatment, high resection rate (79% - 100%), 77% -94% radical operations and 5% - 35% complete histological regressions, it was not
introduced as standard treatment to clinical practice due to lack of published III phase studies. In the literature there are insufficient publications analyzing long term survival after
neoadjuvant gastric cancer treatment. Second phase trials mainly reported as the early stage results with no long term survival analysis. The presented article is of great importance in terms of assessing the effectiveness of neoadjuvant gastric cancer treatment, despite a retrospective analysis.
Minor comments
Fig. 1 Instead of ”Chemoctherapy” should be ”Chemotherapy”
2.2. Radiotherapy – The Authors described the planning of radiotherapy based on the recommendations of the ICRU 50 protocol. It is well known that the stomach and regional
lymph nodes change their positions during breathing. In this situation, it is better to use 62 ICRU protocol which allows to construct the ITV (not only CTV). The Authors did not use
respiratory-gated deep inspiration breath hold (DIBH). Another problem is the CTV-PTV margins. There is no information how the margines were calculated. The 8 mm margins for CTV-PTV in craniocaudal direction may be insufficient for covering respiratory movements of
the stomach.
In the discussion, it is worth discussing the problem of the risk of geographic error that may result from the described treatment planning.
”The short case report” concerning Fig 2. is not necessary.
The Authors repeated the relation between GTV and CTV twice. Once is enough.
3. Results
The Authors presented univariate analysis of treatment related prognostic factors (type of chemotherapy, type of surgery, rate of pCR, down staging). There are no information about analyzed non statistical significant prognostic factors.
The knowledge about negative prognostic factors are important, also. I propose to list all the analyzed prognostic factors. In my opinion, analysis of pretreatment prognostic factors (e.g. weight loss, HGB level and other) may be helpful for the choosing the stratification factors for III phase studies.
4. Discussion
The discussion is very interesting, but not complete. The Authors focused on analysisIII phase studies concerning the esophageal cancer (EC). The principal topic of this
manuscript is a gastric cancer, but not EC. I understand that the Authors looked for an analogy.
It is worth discussing the lacking topic of perioperative chemotherapy, especially thephase III trials: MAGIC and FLOT4-AIO. In Europe according to ESMO guidelines perioperative chemotherapy is a standard treatment. The position of NCCN is ambivalent, it recommends both perioperative chemotherapy as well as preoperative radiochemotherapy. I propose to the authors a comparison of the failure pattern between perioperative chemotherapy and
preoperative radiochemotherapy. I am curious about the Authors' opinion which treatment regimen should be used.
Before the Editorial Board’s final decision, please familiarize yourself with my comments and suggestions. I am happy to reread the manuscript after the corrections have been made.
I should recommend publishing this article in Current Oncology after appropriate corrections.
Author Response
Dear Reviewer:
Thank you for your helpful feedback on improving our current manuscript, we have revised it based on your comments, we hope that these comments have been adequately addressed in an appropriate expression.
We look forward to hearing from you regarding our submission. We would be glad to respond to any further questions and comments that you may have.
Thank you very much.
With kind regards.
Jing Shen.

Reviewer 2 Report
This retrospective study has interesting aspect and it is promising. The paper aims to show that preoperative chemoradiation is effective and feasible in patients with locally advanced resectable gastric cancer.
There are some issues whose resolution will increase the strength of the manuscript and contribute to its publication, as mentioned below:
In the abstract please provide unabbreviated each abbreviation that appears for the first time. In general, Please explain each abbreviation the first time you use it.
P2L68 Please correct “protocal”
Figure 1: Please specify how many did not finish the operation
Figure 2A Please correct “recomanded”
Please explain how patient’s age, physical condition and economic situation affected the choice of chemotherapy regimen. Could these factors influence the results?
Please explain what determined the administration or not of postoperative chemotherapy?
Please rephrase the last sentence of the last paragraph on page 7 as it is not clear.
Please explain the abbreviations in the figure 3 caption.
In table 3, please present the univariate analysis of other factors, such as age, gender, etc.
Please explain which factors were used in the multivariate analysis and with what criteria?
P9L229 Probably you mean large clinical trials. It is interesting that although Stahl M et al conclude that their data suggest a benefit in local progression-free survival when radiotherapy was added to preoperative chemotherapy, they admit that the primary end-point overall survival of the study was not met.
P9L240 It is not clear which clinical studies you are referring to. Please enter the relevant references.
P9L244 Xu, X., et al conclude that pCR is a promising prognostic indicator. Wang, T., et al did not examine the prognostic function of pCR. It is not clear how reference number 21 supports the statement at this point in the article.
P9L247 Please explain what literature supports this conclusion? If it is only reference number 22 then it would be better to note that the Petrellis F et al report these results.
P9L252 Please refer these results to the Results section. How did you analyze the effect on OS of the adjuvant chemotherapy? Could you please provide the univariable Cox regression analysis for this factor? Could this factor used in a multivariable Cox regression analysis?
P9L260 Correct “patient”
P9L255-264 Please move this paragraph to the Results section.
Please refer to the table 4
Author Response

(The authors gave the same response as above.)

Round 2
Reviewer 1 Report
Congratulations.
Author Response
Dear Reviewer:
Thank you for giving us such a precious opportunity, we are glad to hear from you and receive your valuable advice. We have revised the manuscript one by one according to your comments.
1. Corrected the expression of PCR, pathology complete response →pathological complete response.
2. The expressions of PCR are unified, which are all capitalized, and corrected throughout the article.
We look forward to hearing from you regarding our submission. We would be glad to respond to any further questions and comments that you may have.
Thank you very much.
With kind regards.
Jing Shen.
